# Prevalence, Risk Factors, and Impact of Long COVID Among Adults in South Korea

**DOI:** 10.3390/healthcare12202062

**Published:** 2024-10-17

**Authors:** Ha-Eun Son, Young-Seoub Hong, Seungho Lee, Hyunjin Son

**Affiliations:** 1Department of Preventive Medicine, College of Medicine, Dong-A University, Busan 49201, Republic of Korea; haundl@gclabs.co.kr (H.-E.S.); yshong@dau.ac.kr (Y.-S.H.); seunglee@dau.ac.kr (S.L.); 2Infectious Disease Research Center, Green Cross Laboratories, Yongin 16924, Republic of Korea

**Keywords:** post-COVID-19 condition, COVID-19, prevalence, risk factors, symptoms, daily activity limitations

## Abstract

Objectives: This study aimed to identify the prevalence, risk factors, and impact of long COVID in a community-based representative sample of patients with COVID-19 aged 19–64 years. Methods: A total of 975 participants completed online or telephone surveys at 1 and 3 months post-diagnosis, covering persistent symptoms, daily activity limitations, vaccination status, and underlying diseases. Results: Long COVID, as defined by the WHO criteria, had a prevalence of 19.7–24.9% in females and 12.7% in males. Logistic regression revealed that the odds of having long COVID symptoms were higher among females compared to males (OR, 2.43; 95% CI, 1.53–3.87), and higher in those aged ≥ 30 years compared to those aged 19–29 years: 30–39 years (OR, 2.91; 95% CI, 1.59–5.33), 40–49 years (OR, 2.72; 95% CI, 1.51–4.89), and 50–64 years (OR, 1.96; 95% CI, 1.10–3.49). Additionally, patients with underlying diseases had higher odds of long COVID symptoms compared to those without underlying diseases (OR, 1.81; 95% CI, 1.24–2.64). Among those with long COVID, 54.2% experienced daily activity limitations, and 40.6% received treatment. Furthermore, lower income groups faced greater daily activity limitations but had similar treatment rates to higher income groups. Conclusions: These findings emphasize the need for interest in and the development of programs to support these low-income populations.

## 1. Introduction

More and more evidence indicates that patients may experience persistent symptoms beyond the acute phase of COVID-19 [1]. The impact of COVID-19 has been unprecedented, while long-term symptoms may have even more fatal impacts [2]. The appearance of persistent symptoms in previously infected individuals is commonly referred to by several terms, including post-COVID-19 condition, post-acute COVID-19 syndrome, post-acute sequelae of SARS-CoV-2 infection (PASC), and long COVID [1,3,4]. Different countries and organizations use different terms and definitions, as definitions have not been established. The World Health Organization (WHO) defined post-COVID-19 condition (long COVID) as the continuation or development of new symptoms 3 months after the initial SARS-CoV-2 infection, with these symptoms lasting for at least 2 months with no other explanation [5]. The National Institute for Health and Care Excellence (NICE) of the United Kingdom defined ongoing symptomatic COVID-19 as signs and symptoms of COVID-19 persisting 4 to 12 weeks and post-COVID-19 syndrome as signs and symptoms that develop during or after an infection consistent with COVID-19, which continue for more than 12 weeks and are not explained by an alternative diagnosis [3].

Individuals with long COVID exhibit involvement and impairment of the structure and function of multiple organs, with symptoms ranging from mild to severe [6,7,8,9,10]. The typical signs and symptoms, including fatigue, shortness of breath, and cognitive dysfunction, impact daily activity and are non-specific and subjective [10]. Moreover, such persistent symptoms impact various aspects of daily activity, such as physical and cognitive functions, health-related quality of life, and social participation [11].

In June 2023, the WHO released a statement that over 36 million people in Europe were experiencing or had experienced long COVID in the past 3 years [12]. However, an understanding of long COVID is still lacking. Many countries have conducted studies and investigations on various dimensions, including prevalence, pathogenesis, treatment modalities, diagnostics, and long-term effects [13,14]. South Korea must conduct research because the epidemiology of COVID-19 varies widely between countries worldwide, and includes vaccination rates and treatment capacity [15].

Studies conducted in Korea have limitations in analyzing the prevalence and risk factors because such studies have mostly focused on inpatient COVID-19 cases, lacked representative samples, or have been retrospective [16,17,18]. Accordingly, this study aimed to identify the prevalence, risk factors, and impact of long COVID through a prospective investigation using representative community samples.

## 2. Materials and Methods

### 2.1. Study Population

This study used systematic sampling to select 1648 patients (34.5%) among 4779 PCR-positive confirmed patients with COVID-19 aged 19–64 years, reported to a single public health center in Busan during the 5 weeks between December 2022 and January 2023. The selected participants completed a baseline survey at 1 month post-diagnosis through an online or telephone survey. Participants who reported at least one persistent symptom in the 1-month survey were followed up at 3 months post-diagnosis, and an assessment for long COVID symptoms was conducted. Of the 1648 participants, 975 completed the surveys and were included in the final analysis.

### 2.2. Questionnaire Components

The variables for COVID-19-related characteristics consisted of the number of positive COVID-19 cases, vaccination status, time and place of treatment, use of COVID-19 medication, type and persistence of symptoms, and limitation on daily activity because of persistent symptoms. Place of treatment had separate questions for the place of treatment at the time of confirmed COVID-19 diagnosis and the place of treatment for persistent symptoms after release from isolation. The response choices for the questions on place of treatment were “not treated”, “pharmacy only”, “outpatient treatment”, and “inpatient treatment”. The questions on the types and persistence of symptoms were classified based on whether systemic, cardiopulmonary, ophthalmic, gastrointestinal, neurological, neurocognitive, psychological, and musculoskeletal symptoms persisted up to the present. Limitation on daily activity was categorized into “no limitation”, “slight limitation”, “moderate limitation”, “severe limitation”, and “very severe limitation” because of persistent symptoms.

Demographics and health-related characteristics consisted of monthly household income, smoking status, height, weight, and presence of underlying disease. Monthly household income was categorized into KRW < 1 million, KRW 1 to <3 million, KRW 3 to <5 million, and KRW ≥ 5 million. Smoking status was categorized into “never smoked”, “smoked in the past”, and “currently smoking”. The presence and types of underlying disease were based on diagnosis by a physician.

### 2.3. Statistical Analysis

In this study, long COVID was defined as the continuation of symptoms 3 months after the onset of COVID-19 symptoms, with these symptoms lasting for at least 2 months with no other explanation, based on the definition of post-COVID-19 condition (long COVID) by the WHO. The analysis was performed on the long COVID group classified based on this definition [5]. Demographics, COVID-19-related characteristics, and each group’s daily activity limitation were presented as frequencies and percentages. Moreover, the differences between the groups were tested using the chi-square or Fisher’s exact test. The risk factors and impact of long COVID were identified by calculating the odds ratio (OR) and 95% confidence interval (CI) using logistic regression analysis. Moreover, the Cochran–Armitage test and logistic regression analysis were performed to identify the trends and associations between daily activity limitation and whether treatment was received, according to sex, age, and monthly household income. The statistical significance level was set at 0.05 and all analyses were performed using R version 4.2.3.

### 2.4. Ethics Statement

This study was approved by the Institutional Review Board of Dong-A University (IRB No. 2-1040709-AB-N-01-202209-HR-046-10).

## 3. Results

### 3.1. Demographics, Health, and COVID-19-Related Characteristics of Survey Participants

Table 1 presents the results from the analysis of differences in demographics and health/COVID-19-related characteristics according to the presence of long COVID symptoms. The results showed that 19.7% (192/975) of the participants had long COVID symptoms, 24.9% (139/559) among females and 12.7% (53/416) among males. Concerning demographics and health-related factors, significant differences between the long COVID and non-long COVID groups were found based on sex, age, underlying disease, and smoking status. Among males, only age showed a significant difference. Meanwhile, the groups had no significant differences based on COVID-19-related characteristics.

### 3.2. Prevalence by Symptom Cluster at 1 and 3 Months (with Duplicates)

Figure 1 shows the results from the analysis of the types of persistent symptoms found at 1 and 3 months after COVID-19 diagnosis. When symptoms were classified by clusters according to the criteria by the Korean Society of Infectious Diseases [10], the results showed that the most prevalent symptom cluster at 3 months was cardiopulmonary symptoms (12.1%), followed in order by systemic symptoms (8.6%), psychological symptoms (5.0%), neurological symptoms (4.4%), musculoskeletal symptoms (4.1%), and ophthalmic and neurocognitive symptoms (3.8%). Gastrointestinal symptoms (3.1%) were found to be the least prevalent.

### 3.3. Results of Risk Factors Associated with Long COVID

Figure 2 shows the results from the logistic regression analysis of risk factors associated with long COVID. The results showed that the OR of having long COVID symptoms was higher among females as compared to males (OR, 2.43; 95% CI, 1.53–3.87) and among those aged ≥ 30 years as compared to those aged 19–29 years: 30–39 years (OR, 2.91; 95% CI, 1.59–5.33), 40–49 years (OR, 2.72; 95% CI, 1.51–4.89), and 50–64 years (OR, 1.96; 95% CI, 1.10–3.49). Moreover, the OR was higher in patients with underlying disease than in those without (OR, 1.81; 95% CI, 1.24–2.64).

### 3.4. Impact of Long COVID on Daily Activity and Medical Usage

Figure 3 shows the results from analyzing the limitation on daily activity and medical usage because of long COVID. Among the participants, 54.2% reported that long COVID limited their daily activity, while 45.8% reported that their symptoms did not limit their daily life. Moreover, the proportion of individuals receiving treatment for long COVID symptoms based on the degree of daily activity limitation is as follows: 28.4% in the “No limitation on activity” group, 48.5% in the “Slight limitation on activity” group, 56.0% in the “Moderate limitation on activity” group, 55.6% in the “Severe limitation on activity” group, and 50.0% in the “Very severe limitation on activity” group. Furthermore, among the 40.6% of participants who reported receiving treatment for long COVID symptoms, 64.1% reported that they visited a medical institution, whereas 35.9% reported that they only visited a pharmacy. A significant portion (43.6%) of the participants reported receiving treatment for at least 4 weeks.

Table 2 shows the trends and associations of limitation on daily activity and whether treatment was received according to sex, age, and monthly household income. The lowest income group (KRW < 1 million) showed the highest percentage of participants reporting daily activity limitation with 70.4%. The results showed a statistically significant trend with 60.0%, 52.0%, and 31.4% in the KRW 1 to <3 million, KRW 3 to <5 million, and KRW ≥ 5 million groups, respectively. However, the results showed no significant trend in treatment because of persistent symptoms based on household income.

Meanwhile, sex- and age-adjusted logistic regression analysis results showed that the OR of daily life being limited because of persistent symptoms was higher among those with a household income of 1 to <3 million KRW (OR, 3.44; 95% CI, 1.46–8.12) and <1 million KRW (OR, 5.39; 95% CI, 1.77–16.35) than those with ≥5 million KRW. However, the results showed no association between household income and the experience of receiving treatment for long COVID symptoms.

## 4. Discussion

This study used representative community samples to identify the prevalence, risk factors, and the extent of daily activity limitations associated with long COVID.

The results of this study can be summarized into three major findings. Firstly, the prevalence of long COVID among adults at 3 months after COVID-19 infection was 19.7%, with the prevalence being nearly twice as high among females (24.9%) than males (12.7%). Currently, estimates of the prevalence of long COVID vary because of significant differences in major research methodologies, including sample recruitment methods (e.g., hospital, non-hospital, and self-selection), the definition of long COVID, and follow-up periods [19]. Among various studies, a retrospective study in Korea that used Health Insurance Review and Assessment Service data found that 19.1% of patients with COVID-19 experienced at least one sequela, similar to the prevalence found in this study [20]. Moreover, the finding that females have a higher prevalence than males is consistent with previous studies, with a meta-analysis study conducted outside of Korea reporting that the risk of long COVID was twice as high among females than males [21]. Other studies in Korea also confirmed that the risk of neuropsychiatric long COVID was higher among females [18].

Secondly, female sex, age ≥ 30 years, and the presence of underlying disease were identified as risk factors associated with long COVID. Biological and immunological differences between sexes have been discussed as potential explanations for why females experience long COVID more frequently than males [22,23]. Additionally, females tend to experience higher levels of stress, depression, and poorer sleep quality, which may also contribute to the increased incidence of long COVID in females [24]. Moreover, an increase in age is associated with persistent fatigue, musculoskeletal pain, and impaired lung function, which may reflect decreased organ function and a decline in the ability to recover [25,26]. It is also known that the progression of COVID-19 tends to be more severe in patients with an underlying disease, such as hypertension, obesity, and diabetes. Accordingly, the duration of symptoms may also be longer [27,28]. On the other hand, the findings of our study regarding the relationship between vaccination status and long COVID did not show a significant effect. Many studies conclude that COVID-19 vaccination either improves long COVID symptoms or has no significant effect. However, some previous research has mentioned the possibility of worsening or persistent long COVID symptoms following vaccination [29]. These differences in study outcomes may be due to various factors, such as underlying conditions, immune responses, or psychological factors specific to individual patients [30,31]. It is important to note that the relationship between vaccination and long COVID remains complex and is not yet fully understood.

Thirdly, 54.2% of the participants with long COVID reported a limitation on daily activity, with a greater impact among those with lower household incomes. According to community health surveys conducted in Korea, the rates of experiencing limitations in daily activities due to post-COVID-19 symptoms that persisted for over 4 weeks were 16.2% for “severely affected” and 56.3% for “mildly affected” [32], which were higher than the rate of limitation on daily activity found in this study. It is believed that such a difference can be attributed to the fact that this study surveyed the experience of limitation on daily activity based on the continuation of symptoms for more than 12 weeks, based on the definition of long COVID given by the WHO rather than continuation of symptoms for 4 weeks. Persistent symptoms associated with long COVID impacted various aspects of daily activity, including health-related quality of life and participation in society [11,33], indicating the need for responses for preventing and managing such symptoms. In addition, 40.6% of the participants with long COVID reported receiving treatment for persistent symptoms. However, unlike the experience of limitation on daily activity, differences based on household income were not found. Accordingly, social support and interest are believed to be needed more than medical aspects for lower income groups.

The limitations of this study included the fact that older people and children who may have a relatively difficult time participating in online surveys were excluded. Only adults who could participate in the online surveys independently were included. Moreover, the use of a self-reporting format for reporting symptoms instead of objective physiological or cognitive measures was another limitation of this study.

Despite these limitations, this study investigated community prevalence using representative community samples, not just specific patient groups, to conduct a prospective survey. This study is also significant because it used systematic sampling rather than voluntary enrollment. Furthermore, conducting the surveys only at 3 months may cause the participants to experience difficulties recalling their symptoms. However, this study minimized recall bias by conducting a baseline survey at 1 month and a follow-up survey at 3 months.

## 5. Conclusions

This study identified estimates of the actual prevalence and risk factors of long COVID, including mild cases, in a community-based representative sample in Korea. Additionally, more than half of the individuals in the long COVID group experienced limitations in their daily activities due to persistent symptoms, and a significant portion received treatment. It is necessary to have a greater understanding of the etiology, risk factors, symptoms, and treatment modalities for long COVID to reduce the burden and needs of people with diseases and healthcare systems that work to support such patients. Moreover, future studies are also needed to identify pathophysiological mechanisms and develop programs to prevent and treat long COVID within the community population. Furthermore, the findings also showed that lower income groups experienced more limitation on daily activity but received treatment at rates similar to higher income groups. Therefore, there is also the need for interest in and the development of programs to provide social support to low-income groups.

## Figures and Tables

**Figure 1 healthcare-12-02062-f001:**
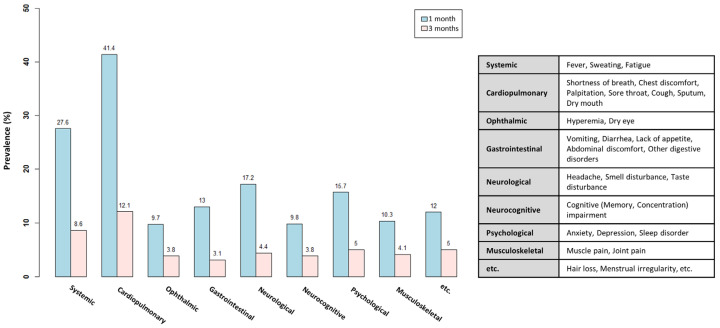
Prevalence by symptom cluster at 1 and 3 months (with duplicates).

**Figure 2 healthcare-12-02062-f002:**
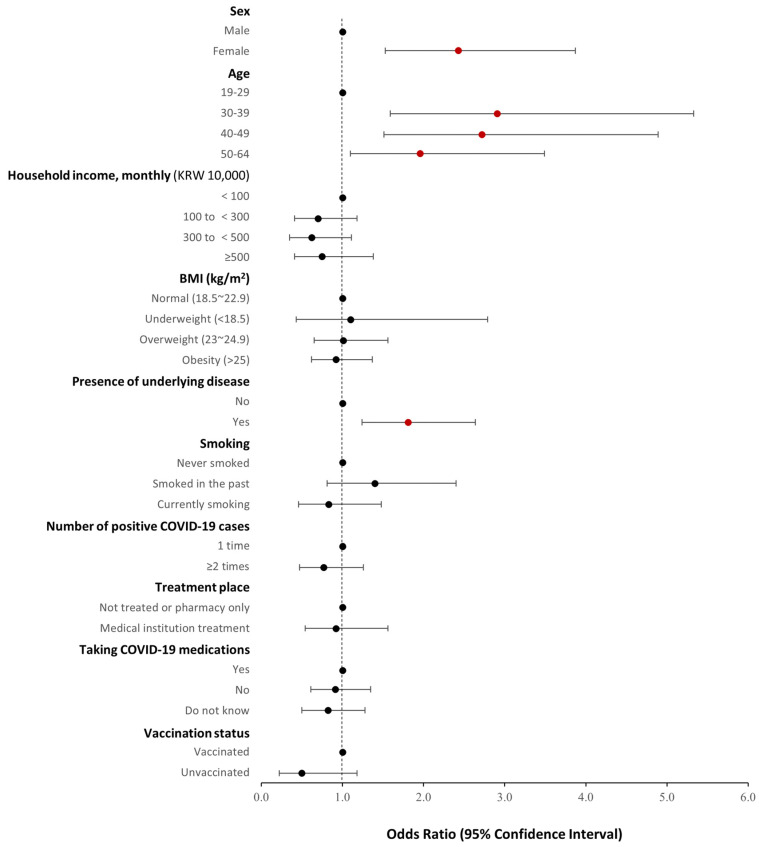
Results of risk factors associated with long COVID.

**Figure 3 healthcare-12-02062-f003:**
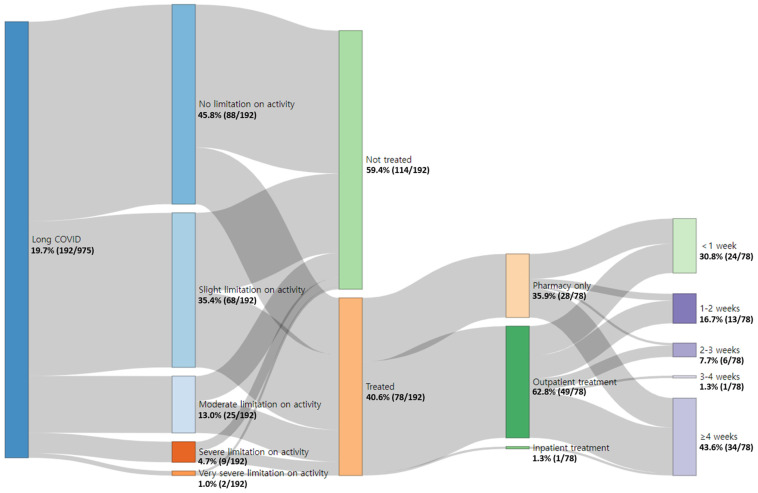
Impact of long COVID on daily activity and medical usage.

**Table 1 healthcare-12-02062-t001:** Demographics, health, and COVID-19-related characteristics of survey participants.

Characteristic	Total	Long COVID	Non-Long COVID	*p*-Value
	N (Column %)	N (%)	N (%)	
**Total**	975	192 (19.7)	783 (80.3)	
**Sex**				<0.001 ^a^
Male	416 (42.7)	53 (12.7)	363 (87.3)	
Female	559 (57.3)	139 (24.9)	420 (75.1)	
**Age (years)**				<0.001 ^a^
19–29	211 (21.6)	20 (9.5)	191 (90.5)	
30–39	205 (21.0)	42 (20.5)	163 (79.5)	
40–49	242 (24.8)	57 (23.6)	185 (76.4)	
50–64	317 (32.5)	73 (23.0)	244 (77.0)	
**Household income, monthly (KRW 10,000)**				0.40 ^a^
<100	106 (10.9)	27 (25.5)	79 (74.5)	
100 to <300	411 (42.2)	80 (19.5)	331 (80.5)	
300 to <500	282 (28.9)	50 (17.7)	232 (82.3)	
≥500	176 (18.1)	35 (19.9)	141 (80.1)	
**BMI (kg/m^2^)**				0.75 ^a^
Underweight (<18.5)	30 (3.1)	7 (23.3)	23 (76.7)	
Normal (18.5~22.9)	358 (36.7)	75 (20.9)	283 (79.1)	
Overweight (23~24.9)	227 (23.3)	45 (19.8)	182 (80.2)	
Obesity (≥25)	355 (36.4)	64 (18.0)	291 (82.0)	
**Presence of underlying conditions**				<0.001 ^a^
No	750 (76.9)	127 (16.9)	623 (83.1)	
Yes	225 (23.1)	65 (28.9)	160 (71.1)	
**Smoking**				0.01 ^a^
Never smoked	636 (65.2)	138 (21.7)	498 (78.3)	
Smoked in the past	154 (15.8)	32 (20.8)	122 (79.2)	
Currently smoking	185 (19.0)	22 (11.9)	163 (88.1)	
**The number of positive COVID-19 cases**				0.55 ^a^
1 time	832 (85.3)	167 (20.1)	665 (79.9)	
≥2 times	143 (14.7)	25 (17.5)	118 (82.5)	
**Treatment place**				0.19 ^b^
Not treated	32 (3.3)	2 (6.3)	30 (93.8)	
Pharmacy-only	113 (11.6)	20 (17.7)	93 (82.3)	
Outpatient treatment	825 (84.6)	169 (20.5)	656 (79.5)	
Inpatient treatment	5 (0.5)	1 (20.0)	4 (80.0)	
**Taking COVID-19 medications**				0.20 ^a^
No	463 (47.5)	96 (20.7)	367 (79.3)	
Yes	265 (27.2)	57 (21.5)	208 (78.5)	
Do not know	247 (25.3)	39 (15.8)	208 (84.2)	
**Vaccination status**				0.27 ^a^
Vaccinated	921 (94.5)	185 (20.1)	736 (79.9)	
Unvaccinated	54 (5.5)	7 (13.0)	47 (87.0)	

^a^ *p*-value of chi-square test; ^b^ *p*-value of Fisher’s exact test.

**Table 2 healthcare-12-02062-t002:** Impact of long COVID on daily activity and medical usage.

Characteristic	Total	Limitation on Daily Activity					Treatment for Persistent Symptoms				
		No	Yes	*p*-Value	Unadjusted	Adjusted ^†^	Not Treated	Treated	*p*-Value	Unadjusted	Adjusted ^†^
	N (Column %)	N (%)	N (%)		OR (95% CI)	OR (95% CI)	N (%)	N (%)		OR (95% CI)	OR (95% CI)
**Total**	192	88 (45.8)	104 (54.2)				114 (59.4)	78 (40.6)			
**Sex**				0.95 ^a^					0.74 ^a^		
Male	53 (27.6)	25 (47.2)	28 (52.8)		Ref	Ref	33 (62.3)	20 (37.7)		Ref	Ref
Female	139 (72.4)	63 (45.3)	76 (54.7)		1.08 (0.57–2.03)	0.96 (0.49–1.90)	81 (58.3)	58 (41.7)		0.85 (0.44–1.62)	0.91 (0.46–1.79)
**Age (years)**				0.68 ^b^					0.12 ^b^		
19–29	20 (10.4)	10 (50.0)	10 (50.0)		Ref	Ref	14 (70.0)	6 (30.0)		Ref	Ref
30–39	42 (21.9)	20 (47.6)	22 (52.4)		1.10 (0.38–3.19)	1.02 (0.33–3.14)	26 (61.9)	16 (38.1)		0.70 (0.22–2.18)	0.69 (0.22–2.20)
40–49	57 (29.7)	25 (43.9)	32 (56.1)		1.28 (0.46–3.55)	1.40 (0.47–4.19)	36 (63.2)	21 (36.8)		0.73 (0.25–2.20)	0.69 (0.22–2.14)
50–64	73 (38.0)	33 (45.2)	40 (54.8)		1.21 (0.45–3.26)	1.07 (0.38–3.03)	38 (52.1)	35 (47.9)		0.47 (0.16–1.34)	0.46 (0.16–1.35)
**Household income, monthly**											
(KRW 10,000)				0.001 ^b^					0.56 ^b^		
≥500	35 (18.2)	24 (68.6)	11 (31.4)		Ref	Ref	22 (62.9)	13 (37.1)		Ref	Ref
300 to <500	50 (26.0)	24 (48.0)	26 (52.0)		2.36 (0.96–5.84)	2.27 (0.91–5.68)	31 (62.0)	19 (38.0)		0.96 (0.40–2.35)	1.01 (0.41–2.51)
100 to <300	80 (41.7)	32 (40.0)	48 (60.0)		3.27 (1.41–7.60)	3.44 (1.46–8.12)	45 (56.3)	35 (43.8)		0.76 (0.34–1.72)	0.80 (0.35–1.84)
<100	27 (14.1)	8 (29.6)	19 (70.4)		5.18 (1.74–15.44)	5.39 (1.77–16.35)	16 (59.3)	11 (40.7)		0.86 (0.31–2.41)	0.98 (0.34–2.82)

^a^ *p*-value of chi-square test; ^b^ *p*-value of Cochran–Armitage test; OR = odds ratio, CI = confidence interval; ^†^ adjusted sex and age.

## Data Availability

The data that support the findings of this study are available from the corresponding author upon reasonable request.

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
