# Peer review of "Prevalence, Risk Factors, and Impact of Long COVID Among Adults in South Korea"

_healthcare, 2024, doi:10.3390/healthcare12202062_

Round 1

Reviewer 1 Report

Comments and Suggestions for Authors

The manuscript is scientifically not sound. Patient selection for long COVID is not straightforward. According to the official official definition (WHO, NICE), patients were selected too early (at 1 and 3 months post-diagnosis). There are no detailed information for the exclusion of an alternative diagnosises. These fundamentally determines the negative justification for the research. A thorough, repeated transcription of the manuscript is essential.

Author Response

Reviewer #1

  1. The manuscript is scientifically not sound. Patient selection for long COVID is not straightforward. According to the official official definition (WHO, NICE), patients were selected too early (at 1 and 3 months post-diagnosis). There are no detailed information for the exclusion of an alternative diagnosises. These fundamentally determines the negative justification for the research. A thorough, repeated transcription of the manuscript is essential.

Answer: Thank you for your valuable feedback. We understand your concern regarding patient selection for long COVID. In our study, we followed the WHO's official definition of long COVID, which classifies it as the continuation of symptoms 3 months after the onset of COVID-19 symptoms, with these symptoms lasting for at least 2 months with no other explanation. To minimize recall bias that could occur if only assessing patients at 3 months post-diagnosis, we conducted a baseline survey at 1 month post-diagnosis in PCR positive confirmed case COVID-19 cases reported to a public health center. (The survey at 1 month was conducted for baseline purposes and not for diagnosing long COVID) Participants who reported at least one persistent symptom at 1 month were followed up at 3 months, and only those with symptoms persisting beyond 3 months, in accordance with WHO guidelines, were classified into the long COVID group. Based on your suggestion, we have revised the Methods section to clarify this process and avoid any potential confusion. The revised content is as follows.

Lines 61~67

This study used systematic sampling to select 1,648 patients (34.5%) among 4,779 PCR positive confirmed patients with COVID-19 aged 19–64 years reported to a single public health center in Busan during 5 weeks between December 2022 and January 2023. The selected participants completed a baseline survey at 1 month post-diagnosis through online or telephone survey. Participants who reported at least one persistent symptom in the 1 month survey were followed up at 3 months post-diagnosis, and an assessment for long COVID symptoms was conducted.

Reviewer 2 Report

Comments and Suggestions for Authors

A nice and interesting study. I do have a couple of suggestions and comments to make on it. 

Were you able to ask or gauge the actual severity of the symptoms from those who reported having Long COVID-19?

You did an adequate job of reporting the prevalence by symptom cluster. Also, I would be wondering the severity of the symptoms from those suffering Long COVID-19, perceived low-severity vs. high-severity.

Also, did you look at perceived current health status? I know you asked about underlying conditions, but how do the participants rate their current health status? How about previous infection of COVID-19? Some patients do get multiple bouts of infection.

Author Response

Reviewer #2

  1. Were you able to ask or gauge the actual severity of the symptoms from those who reported having Long COVID-19?

Answer: Thank you for your question. Yes, in this study, we followed up with participants who reported persistent symptoms at the 1-month mark. At 3 months, we conducted telephone questionnaire surveys asking whether those symptoms were still present. If the symptoms persisted, we asked participants to rate the severity of their symptoms based on their perception, choosing from the following categories: 'very mild,' 'mild,' 'moderate,' 'severe,' or 'very severe.' This allowed us to gauge the severity of their symptoms.

  1. You did an adequate job of reporting the prevalence by symptom cluster. Also, I would be wondering the severity of the symptoms from those suffering Long COVID-19, perceived low-severity vs. high-severity.

Answer: Thank you for your insightful question. In this study, we did collect data on symptoms severity. However, as the focus of this manuscript is on prevalence, risk factors, and impacts, we had to exclude the symptom severity data due the limited scope of the manuscript. We will consider including this information in a future study.

  1. Did you look at perceived current health status? I know you asked about underlying conditions, but how do the participants rate their current health status? How about previous infection of COVID-19? Some patients do get multiple bouts of infection.

Answer: Thank you for your question. As you mentioned, we collected data on underlying conditions that participants had at the time of their COVID-19 diagnosis. Additionally, we conducted a survey asking whether participants believed their underlying conditions worsened after their COVID-19 diagnosis. Responses were categorized as: 'Not worsened at all,' 'Slightly worsened,' 'Moderately worsened,' 'Significantly worsened,' and 'Severely worsened.' Overall, the results were as follows. Among participants with pre-existing conditions, approximately 69% (155/225) reported that their conditions had 'Not worsened at all' following their COVID-19 diagnosis, while about 4% (9/225) reported 'Significantly worsened,' the lowest percentage.

We also surveyed participants on how many times they had tested positive for COVID-19. The results are presented in Table 1 of the manuscript under "The number of positive COVID-19 cases," and the values are as follows. About 85% (832/975) reported having had COVID-19 only once, while approximately 15% (143/975) reported testing positive two or more times.

Reviewer 3 Report

Comments and Suggestions for Authors

The finding regarding long COVID and vaccination status is interesting.

Most journal papers report reduced risk of long COVID for those vaccinated.

However, this article found the opposite

How does COVID-19 vaccination affect long-COVID symptoms?

Asadi-Pooya AA, Nemati M, Shahisavandi M, Nemati H, Karimi A, Jafari A, Nasiri S, Mohammadi SS, Rahimian Z, Bayat H, Akbari A, Emami A, Eilami O.PLoS One. 2024 Feb 7;19(2):e0296680. doi: 10.1371/journal.pone.0296680. 

In my opinion, it seems likely that vaccination would increase the risk of long COVID, especially since the rate of excess deaths increased in many countries after widespread vaccination starting in 2021, but not during the year prior. Please discuss. It could be the fact that studies reporting no effect of vaccinations were incorrect.

Secondly, female sex, age ≥30 years, and presence of underlying disease were identi-

187

fied as risk factors associated with long COVID. Biological (e.g., hormonal and immune

188

responses) and sociocultural (e.g., hygiene-related behaviors, psychological stress, inac-

189

tivity, etc.) aspects play an important role in creating gender-based differences in long

190

COVID symptoms [22], and it is believed that females may be more vulnerable to such

191

factors.

22. Gebhard, C.E.; Sütsch, C.; Bengs, S.; Todorov, A.; Deforth, M.; Buehler, K.P.; et al. Understanding the impact of sociocultural 304 gender on post-acute sequelae of COVID-19: a Bayesian approach. medRxiv 2021, 2021.06.30.21259757

Comment: It is rarely acceptable to cite non-peer reviewed manuscripts. Suggest finding better references for this point and discussing in more detail.

A search of Google Scholar and pubmed.gov finds that low serum 25(OH)D concentration is associated with increased risk of long COVID

Low vitamin levels are associated with long COVID syndrome in COVID-19 survivors

L di FilippoS Frara, F Nannipieri… - The Journal of …, 2023 - academic.oup.com

A Critical Review on the Long-Term COVID-19 Impacts on Patients With Diabetes.

Ashique S, Mishra N, Garg A, Garg S, Farid A, Rai S, Gupta G, Dua K, Paudel KR, Taghizadeh-Hesary F.Am J Med. 2024 Mar 12:S0002-9343(24)00133-5. doi: 10.1016/j.amjmed.2024.02.029.

Prevalence and characteristics of long COVID-19 in Jordan: A cross sectional survey.

Obeidat M, Abu Zahra A, Alsattari F.PLoS One. 2024 Jan 26;19(1):e0295969. doi: 10.1371/journal.pone.0295969.

Do vitamin D levels or supplementation play A role in COVID-19 outcomes?-a narrative review.

Shetty AJ, Banerjee M, Prasad TN, Bhadada SK, Pal R.Ann Palliat Med. 2024 Jan;13(1):162-177. doi: 10.21037/apm-23-113. 

Comment; No data are probably available for the participants in this study and even if they were they may not be for the period just prior to development of COVID-19. However, the comorbid diseases are known and some if not all of them are associated with vitamin D deficiency. Please give more information about the comorbid diseases and whether this information supports the role of higher vitamin D level in reducing risk of long COVID.

Significant digits. The general rule is that no more non-zero digits should be given than are justified by the uncertainty of the value.

See "Too many digits: the presentation of numerical data"

https://www.ncbi.nlm.nih.gov/pmc/articles/PMC4483789/

If the uncertainty (or difference when comparing numbers) is greater than about 7%, only two non-zero digits are justified.

P values should be given to two decimal places unless the first two are 00 or the number lies between 0.045 and 0.054. If the first two are 00, then only one non-zero digit can be given.

Thus, p values should be adjusted.

Please review all numbers in abstract, text, tables, and figures and adjust accordingly.

Author Response

Reviewer #3

  1. The finding regarding long COVID and vaccination status is interesting. Most journal papers report reduced risk of long COVID for those vaccinated. In my opinion, it seems likely that vaccination would increase the risk of long COVID, especially since the rate of excess deaths increased in many countries after widespread vaccination starting in 2021, but not during the year prior. Please discuss. It could be the fact that studies reporting no effect of vaccinations were incorrect.

Answer: Thank you for your insightful feedback. We have added a discussion on the relationship between vaccination status and long COVID in the discussion section of the manuscript. The content is as follows.

Lines 200~208

On the other hand, the findings of our study regarding the relationship between vaccination status and long COVID did not show a significant effect. Many studies conclude that COVID-19 vaccination either improves long COVID symptoms or has no significant effect. However, some previous research has mentioned the possibility of worsening or persistent long COVID symptoms following vaccination [29]. These differences in study outcomes may be due to various factors, such as underlying conditions, immune responses, or psychological factors specific to individual patients [30, 31]. It is important to note that the relationship between vaccination and long COVID remains complex and not yet fully understood.

  1. It is rarely acceptable to cite non-peer-reviewed manuscripts. Suggest finding better references for the discussion on gender-based differences in long COVID.

Answer: Thank you for your valuable feedback. We have carefully reviewed the references and replaced the non-peer-reviewed manuscripts with peer-reviewed sources. The discussion has been revised to include more credible references. The revised sections are as follows.

Lines 191~195

Biological and immunological differences between sexes have been discussed as potential explanations for why females experience long COVID more frequently than males [22, 23]. Additionally, females tend to experience higher levels of stress, depression, and poorer sleep quality, which may also contribute to the increased incidence of long COVID in females [24].

Lines 316~321

22. Bwire, G. M. Coronavirus: why men are more vulnerable to Covid-19 than women? SN Compr Clin Med 2020, 2, 874-876.

23. Ortona, E.; Buonsenso, D.; Carfi, A.; Malorni, W.; Long Covid Kids study group Munblit, D.; De Rose, C.; Sinatti, D.; Ricchiuto, A.; Valentini, P. Long COVID: an estrogen-associated autoimmune disease? Cell Death Discov 2021, 7, 77.

24. Fernández-de-Las-Peñas, C.; Martín-Guerrero, J. D.; Pellicer-Valero, Ó. J.; Navarro-Pardo, E.; Gómez-Mayordomo, V.; Cuadrado, M. L.; Arias-Navalón, J. A.; et al. Female sex is a risk factor associated with long-term post-COVID related-symptoms but not with COVID-19 symptoms: the LONG-COVID-EXP-CM multicenter study. J Clin Med 2022, 11, 413.

  1. No data are probably available for the participants in this study and even if they were they may not be for the period just prior to development of COVID-19. However, the comorbid diseases are known and some if not all of them are associated with vitamin D deficiency. Please give more information about the comorbid diseases and whether this information supports the role of higher vitamin D level in reducing risk of long COVID.

Answer: Thank you for your insightful comments and for highlighting relevant prior studies. In this study, we did not specifically examine the role of vitamin D or its association with comorbid diseases in relation to long COVID. The comorbid diseases in our study include [list of comorbidities, e.g., diabetes, hypertension, cardiovascular diseases, etc.], some of which have been linked to vitamin D deficiency in previous studies. While we did not collect data on participants' vitamin D levels, we agree that this is an important area for future research. Further investigation into the potential role of vitamin D in modulating the risk of long COVID, particularly in populations with specific comorbidities, could provide valuable insights.

  1. Significant digits. The general rule is that no more non-zero digits should be given than are justified by the uncertainty of the value.

See "Too many digits: the presentation of numerical data"

https://www.ncbi.nlm.nih.gov/pmc/articles/PMC4483789/

If the uncertainty (or difference when comparing numbers) is greater than about 7%, only two non-zero digits are justified.

P values should be given to two decimal places unless the first two are 00 or the number lies between 0.045 and 0.054. If the first two are 00, then only one non-zero digit can be given.

Thus, p values should be adjusted.

Please review all numbers in abstract, text, tables, and figures and adjust accordingly.

Answer: Thank you for your valuable comments regarding the presentation of numerical data. We have reviewed all the numbers in the abstract, text, tables, and figures, and have revised some of the p-values in Table 1 and Table 2 of the manuscript in accordance with your suggestions, highlighting the changes in red. We appreciate your guidance in improving the accuracy and clarity of our manuscript.

Round 2

Reviewer 3 Report

Comments and Suggestions for Authors

No further comments.